# ARE IMAGES INDISTINGUISHABLE TO HUMANS ALSO INDISTINGUISHABLE TO CLASSIFIERS?

## ABSTRACT

The ultimate goal of generative models is to perfectly capture the data distribution. For image generation, common metrics of visual quality (e.g., FID) and the perceived truthfulness of generated images seem to suggest that we are nearing this goal. However, through *distribution classification* tasks, we reveal that, from the perspective of neural network-based classifiers, even advanced diffusion models are still far from this goal. Specifically, classifiers are able to consistently and effortlessly distinguish real images from generated ones across various settings. Moreover, we uncover an intriguing discrepancy: classifiers can easily differentiate between diffusion models with comparable performance (e.g., U-ViT-H vs. DiT-XL), but struggle to distinguish between models within the same family but of different scales (e.g., EDM2-XS vs. EDM2-XXL). Our methodology carries several important implications. First, it naturally serves as a diagnostic tool for diffusion models by analyzing specific features of generated data. Second, it sheds light on the model autophagy disorder and offers insights into the use of generated data: augmenting real data with generated data is more effective than replacing it.

## 1 INTRODUCTION

Diffusion probabilistic models (Sohl-Dickstein et al., 2015; Song et al., 2020; Ho et al., 2020) have emerged as leading generative models for image (Ramesh et al., 2021; Esser et al., 2024), video (Brooks et al., 2024; Bao et al., 2024; Zhao et al., 2024) and 3D content generation (Poole et al., 2022), surpassing previous generative models such as generative adversarial networks (Goodfellow et al., 2014), variational autoencoders (Kingma & Welling, 2013), and normalizing flows (Rezende & Mohamed, 2015). In particular, focusing on image generation, diffusion models are capable of producing realistic images that are often indistinguishable from real ones to the human eye (see Fig. 1). Furthermore, common metrics such as FID (Heusel et al., 2017) suggest that the distribution generated by state-of-the-art diffusion models closely approximates the ImageNet validation set (Karras et al., 2023; 2022; Tian et al., 2024). Therefore some works, such as Wang et al. (2024); You et al. (2024), have utilized generated data to augment training datasets. However, to the best of our knowledge, few studies have been conducted to analyze the differences between the generated and real distributions.

This raises a fundamental question: *How far is the generated distribution from the real distribution?* Inspired by the methodology in the work (Liu & He, 2024; Tommasi et al., 2017) on dataset bias (Torralba & Efros, 2011), we propose distribution classification tasks (see Sec. 3 for details), predicting the distribution an image belongs to - real or generated - thereby serving as a probe to explore this critical question. Notably, binary classification is a natural approach, as classifiers parameterized by neural networks can serve as effective indicators of the differences between the real and generated distributions through classification accuracy (see more analysis in Appendix D). Compared to the widely used FID metric, our classifier-based approach directly measures distribution differences without relying on Gaussian assumptions (see further discussion in Appendix E).

We first explore *are images indistinguishable to humans also indistinguishable to classifiers?* To this end, we train several commonly used classifiers to distinguish between the real and generated distribution across various settings. Quantitatively, most classifiers can achieve over 98% accuracy. Meanwhile, all classifiers reach extremely high accuracy with significantly fewer training samples than the requirement of training diffusion models. Interestingly, even self-supervised classifiers exhibit this capacity to identify the generated distribution. These results demonstrate that from the

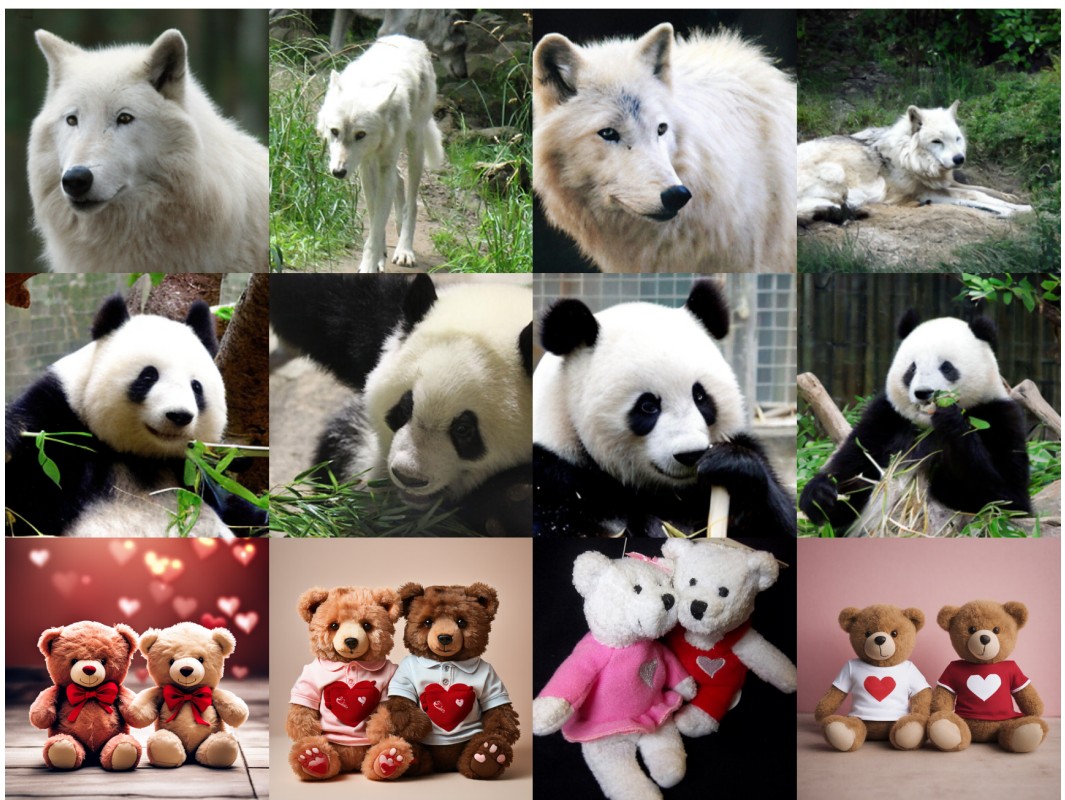

Figure 1: **Four-way distribution classification tasks:** *Which one is real in each row?*[1] The samples are from real images or generated from state-of-the-art diffusion models. Notably, classifiers consistently and effortlessly distinguish between real and generated images in all settings.

perspective of neural network-based classifiers, the generated distribution remains markedly different from the real one, indicating that diffusion models still have significant room for improvement.

Furthermore, we observe intriguing contradictions between classifiers' prediction and both FID and human judgments through two groups of experiments. The first group involves classifying the smallest and largest models within the same diffusion family, which share inductive biases but differ in visual quality. The second group examines diffusion models with similar FID scores and visual quality but different inductive biases. Surprisingly, classifiers perform well on the second task but struggle with the first, revealing contradictions between classifier performance and FID scores. Meanwhile, human judgments align with classifiers in the first task but degrade to random guessing in the second, indicating inconsistency between classifier and human judgments.

Our approach displays several important implications. First, it naturally serves as a diagnostic tool for diffusion models by extending the analysis to specific features of generated data. For example, in evaluating the advanced diffusion transformer U-ViT-H/2 (Bao et al., 2023), we focus on two key aspects: spatial features and frequency ranges. Spatially, classifiers can easily distinguish whether an image is real or generated using any part of the image. In terms of frequency, classifiers struggle with minimal low-frequency information but perform well across all other frequency intervals. This suggests that U-ViT-H/2 has effectively learned low-frequency features, making generated images harder to distinguish in these regions, but remains less proficient in higher frequency regions. Second, our approach enables the evaluation of using generated data for training models. It provides a reasonable explanation for the model autophagy disorder (MAD) phenomenon observed in Shumailov et al. (2024); Alemohammad et al. (2024). This collapse is surprising because, despite the realistic appearance of the generated data, using it in a continuous loop to train subsequent generations of

---

[1] From left to right, the first two rows show images from U-ViT, ImageNet (real), DiT, and EDM2-XXL. The bottom row shows images from Pixart-$\alpha$, Playground-v2.5, COCO (real), and SDXL.

models leads to failure. We explain this by showing that there exists a distribution mismatch between the training and generated distributions in each generation. As this mismatch accumulates, the generated data distribution drifts further from the original. In downstream tasks, such as supervised and semi-supervised classification, this mismatch leads to performance degradation when real data is simply replaced by generated data. However, this distribution drift can be mitigated by *augmenting* real data with generated data, which introduces novel features while maintaining alignment with the original data distribution, leading to improved performance.

Briefly, our key contributions are as follows:

- We show that classifiers easily distinguish between diffusion-generated and real distributions, despite diffusion models achieving low FID scores and generating lifelike images (see Fig. 1).

- We reveal the intriguing contradictions between classifier performance and widely used evaluation metrics, such as FID and human judgments.

- We demonstrate that our approach complements traditional metrics like FID and can serve as a diagnostic tool to provide deeper insights into diffusion models.

- We provide a reasonable explanation for the surprising model autophagy disorder phenomenon and show that augmenting real data with generated data is more effective than replacing it in supervised and semi-supervised learning.

## 2 BACKGROUND

### 2.1 DIFFUSION MODEL

Diffusion models (Sohl-Dickstein et al., 2015; Song et al., 2020; Ho et al., 2020) gradually inject noise into data $x$ during the forward process:

$$z_t = \sqrt{\bar{\alpha}_t} x + \sqrt{1 - \bar{\alpha}_t} \epsilon, \tag{1}$$

and remove noise to generate data in the reverse process. Diffusion models typically use a noise prediction network $\epsilon_\theta(z_t, t)$ to predict the noise $\epsilon$ added to $z_t$. Noise prediction loss is defined as:

$$\mathcal{L} = \mathbb{E}_{t, x_0, \epsilon}[\|\epsilon_\theta(z_t, t) - \epsilon\|_2^2]. \tag{2}$$

### 2.2 FREQUENCY FILTER

The frequency content of an image represents the rate of pixel value changes. Low-frequency components capture overall shapes and gradual grayscale changes, while high-frequency components reflect fine details like edges and textures. Notably, convolutional neural networks can detect high-frequencies that are often imperceptible to humans (Wang et al., 2020a). In our paper, we implement a rectangular filter and discuss the choice between using rectangular or circular filters in Appendix F.

**Low-pass filters** are implemented as follows: Compute the Fourier transform of the image $f(x, y)$ using $F(u, v) = \text{FFT}(f(x, y))$, and then center the spectrum with $F_c(u, v) = \text{fftshift}(F(u, v))$. Define a rectangular mask $H(u, v)$ in the frequency domain, where $M$ and $N$ are the image dimensions:

$$H(u, v) = \begin{cases} 1 & \text{if } \left|u - \frac{M}{2}\right| \leq \text{threshold and } \left|v - \frac{N}{2}\right| \leq \text{threshold,} \\ 0 & \text{otherwise.} \end{cases} \tag{3}$$

Apply the mask by multiplying $G(u, v) = F_c(u, v) \odot H(u, v)$, reverse the shift with $G_c(u, v) = \text{ifftshift}(G(u, v))$, and transform back to the spatial domain using $g(x, y) = \text{IFFT}(G_c(u, v))$.

**High-pass filters** are implemented similarly, but with the mask defined to pass high frequencies:

$$H(u, v) = \begin{cases} 0 & \text{if } \left|u - \frac{M}{2}\right| \leq \text{threshold and } \left|v - \frac{N}{2}\right| \leq \text{threshold,} \\ 1 & \text{otherwise.} \end{cases} \tag{4}$$

**Band-pass filters** combine low-pass and high-pass filters, allowing frequencies between low and high thresholds to pass while setting others to 0.

## 3 RELATED WORK

**Dataset bias.** Torralba & Efros (2011) introduced the "Name That Dataset" game to discover dataset bias. Tommasi et al. (2017) and Liu & He (2024) expanded on this by applying convolutional neural networks and large-scale datasets, respectively. Existing work on dataset bias focuses on classification to highlight biases in data collection for building unbiased datasets, our work leverages distribution classification to explore a fundamental question: *How far is the generated distribution from the real distribution?* Besides, we focus on the discrepancy between the generated distribution and the real distribution rather than between specific datasets. Furthermore, the conclusion drawn by related work highlights the ability of neural networks to detect biases within increasingly general datasets, our conclusion underscores the significant gap that still exists between current diffusion models and the real data distribution. We also highlight the potential of distribution classification methods for gaining deeper insights into generative models and more effectively leveraging generated data.

**Generated image detection.** Several empirical works on generated image detection span three key areas: universal detection (Liu et al., 2024; Ojha et al., 2023; Wang et al., 2020b; Girish et al., 2021), reconstruction-based methods (Wang et al., 2023b; Luo et al., 2024), and frequency analysis (Wang et al., 2020a; Zhang et al., 2019; Durall et al., 2020; Chandrasegaran et al., 2021; Frank et al., 2020; Dzanic et al., 2020; Yang et al., 2023; Corvi et al., 2023b; Ricker et al., 2022; Corvi et al., 2023a). Existing work generally focuses on either building universal discriminators to detect generated images or identifying frequency artifacts to build classifiers. In contrast, our approach leverages distribution classification to gain deeper insights into generative models and quantify the discrepancy between generated and real distributions. Instead of prioritizing classifier performance, we focus on the insights provided into generative model behavior.

## 4 EXPERIMENTAL SETTINGS

We present the main experimental settings as follows. For more details, please see Appendix A.

**Dataset.** For label-to-image tasks, we use the CIFAR-10 (Krizhevsky et al., 2009) and ImageNet (Deng et al., 2009) datasets, and for text-to-image tasks, we utilize the COCO2014 dataset (Lin et al., 2014). For CIFAR-10, the real distribution is constructed from its training and testing sets, comprising 50k training images and 10k testing images. The generated distribution is created using diffusion models to produce an equivalent number of images. For ImageNet, we consider resolutions of 256 and 512 and follow ADM's (Dhariwal & Nichol, 2021) method to preprocess ImageNet into the desired resolutions. We randomly sample 100k training images and 50k validation images from the corresponding ImageNet sets to represent the real distribution, and generate an equivalent number of images using diffusion models for the generated distribution. For COCO2014, we randomly sample 10k training images and 1k validation images from the respective sets to form the real distribution, and use the corresponding prompts to generate an equivalent number of data for the generated distribution.

**Classifier.** By default, we use ResNet-50 (He et al., 2016) as the classifier architecture, and also consider ConvNeXt-T (Liu et al., 2022) and ViT-S (Dosovitskiy et al., 2020) for completeness. Our pre-processing follows standard supervised training (Liu et al., 2022). Specifically, during training, the classifiers process randomly augmented crops of $224 \times 224$ images. For validation, images are resized to 256 pixels on the shorter side, preserving the aspect ratio, and then center cropped to $224 \times 224$. For CIFAR-10 experiments, we initialize the classifier with ResNet-50 pre-trained on ImageNet and also provide the results trained from scratch (see Appendix C.2). For ImageNet, we train ResNet-50 from scratch.

**Diffusion model.** For CIFAR-10 generation, we consider two diffusion models: EDM (Karras et al., 2022) and U-ViT (Bao et al., 2023). For ImageNet-256, we explore three diffusion models: EDM2 (Karras et al., 2023), U-ViT-H/2 (Bao et al., 2023), and DiT-XL/2 (Peebles & Xie, 2023). Since EDM2 is originally designed for ImageNet-512 generation, we resize its generated images from 512 to 256 resolution, with EDM2-XXL achieving an FID of 2.14, comparable to U-ViT-H/2 and DiT-XL/2. For ImageNet-512 generation, we use EDM2-XXL (Karras et al., 2023). For COCO generation, we consider three text-to-image diffusion models: Pixart-$\alpha$ (Chen et al., 2023), SDXL (Podell et al., 2023), and Playground-v2.5 (Li et al., 2024).

**Evaluation.** We use the top-1 accuracy on the validation set to evaluate classification performance.

Table 1: **Binary distribution classification on *the same distribution***. Classifiers are indeed unable to distinguish samples from the same distribution, initially demonstrating the validity of our approach.

| Distribution1 | Distribution2 | Classifier | Accuracy (%) |
|---|---|---|---|
| *U* (Bao et al., 2023) | *U* | ResNet-50 | 50.83 |
| | | ViT-S | 50.57 |
| | | ConvNeXt-T | 50.18 |
| *E* (Karras et al., 2022) | *E* | ResNet-50 | 50.67 |
| | | ViT-S | 50.44 |
| | | ConvNeXt-T | 50.14 |

**Abbreviations.** In the figures and tables following this paper, the following abbreviations are used: CIFAR-10 (*C*), ImageNet (*I*), U-ViT (*U*), EDM (*E*), DiT-XL/2 (*D*).

## 5 GENERATED DISTRIBUTIONS ARE EASILY CLASSIFIED AS GENERATED

In this section, we explore how the distribution generated by diffusion models, known for their success in image generation, differs from the real distribution. To analyze this, we use a distribution classification task to assess the extent of the difference.

### 5.1 CLASSIFIERS CANNOT DISTINGUISH SAMPLES FROM THE SAME DISTRIBUTION

First, it is crucial to evaluate whether classifiers are indeed unable to distinguish samples from the same distribution. As shown in Tab. 1, we report the classification accuracy for the same distributions in the label-to-image scenario. Our findings indicate that classifiers are unable to distinguish samples from the same distribution, initially demonstrating the validity of the distribution classification task.

### 5.2 CLASSIFIERS ACHIEVE HIGH ACCURACY ACROSS VARIOUS SETTINGS

**Label-to-image.** As shown in Tab. 2, we report the binary distribution classification accuracy for various combinations of real and generated distributions in the label-to-image scenario. Surprisingly, despite the low FID scores achieved by diffusion models, neural network-based classifiers still identify significant differences between the generated and real distributions. Specifically, all classifiers easily distinguish real from generated images across different dataset combinations, with most achieving over 98% accuracy in distribution classification. Moreover, we observe a positive correlation between distribution classification accuracy and the number of training samples for the combination of U-ViT-H/2 and ImageNet-256 (see Fig. 2). This suggests that as sample size increases, classifiers learn more generalizable features rather than merely memorizing the data. Notably, with only 50k samples per distribution (less than 10% of the ImageNet samples, approximately 1.28 million, used to train the generator), classifiers achieve over 99.5% accuracy. Besides, we extend our evaluation to other generative models, e.g., GANs (see detailed results in Sec. C.1).

**Text-to-image.** As shown in Tab. 3, we report the distribution classification accuracy in the text-to-image scenario. We consider combinations of four distributions: COCO (Lin et al., 2014), Pixart-$\alpha$ (Chen et al., 2023), SDXL (Podell et al., 2023), and Playground-v2.5 (Li et al., 2024). In this task, classifiers are not only required to distinguish between real and generated distributions but also correctly identify the specific generated distribution an image comes from. Remarkably, classifiers achieve an accuracy of over 76% across all settings. Furthermore, we observe that increasing the training sample size from 5k to 10k results in significant accuracy gains, consistent with the patterns seen in the label-to-image scenario.

### 5.3 SELF-SUPERVISED CLASSIFIERS CAN ALSO IDENTIFY GENERATED IMAGES

In addition to the supervised learning protocol, we also explore self-supervised learning. Specifically, we use self-supervised pre-trained models, such as MAE (He et al., 2022) and MoCo v3 (Chen et al., 2021), to extract image features followed by training a linear classifier on these features, referred to as the self-supervised classifier. The distribution classification accuracy of these classifiers is in

Table 2: **Binary distribution classification on *label-to-image***. All classifiers yield high accuracy on various combinations of datasets and generative models. FIDs are taken from related references.

| Real dataset | Generative model | Corresponding FID | Classifier | Accuracy (%) |
|---|---|---|---|---|
| *C* | *U* (Bao et al., 2023) | 3.11 | ResNet-50 | 99.92 |
| | | | ViT-S | 98.04 |
| | | | ConvNeXt-T | 99.96 |
| *C* | *E* (Karras et al., 2022) | 1.79 | ResNet-50 | 96.25 |
| | | | ViT-S | 89.38 |
| | | | ConvNeXt-T | 98.43 |
| *I*-256 | *U*-H/2 (Bao et al., 2023) | 2.29 | ResNet-50 | 99.95 |
| | | | ViT-S | 98.06 |
| | | | ConvNeXt-T | 99.89 |
| *I*-512 | *E*2-XXL (Karras et al., 2023) | 1.81 | ResNet-50 | 99.73 |
| | | | ViT-S | 95.15 |
| | | | ConvNeXt-T | 86.44 |

Table 3: **Four-way distribution classification on *text-to-image***. All classifiers yield high accuracy in distinguishing four distributions: COCO (Lin et al., 2014), Pixart-$\alpha$ (Chen et al., 2023), SDXL (Podell et al., 2023), and Playground-v2.5 (Li et al., 2024), using only 5k or 10k training samples per dataset. Notably, as the number of training samples increases, the accuracy of distribution classification consistently improves.

| Training samples | Model | Accuracy (%) |
|---|---|---|
| 5k | ResNet-50 | 88.28 |
| 10k | ResNet-50 | 93.45 |
| 5k | ViT-S | 76.33 |
| 10k | ViT-S | 83.53 |
| 5k | ConvNeXt-T | 80.53 |
| 10k | ConvNeXt-T | 85.13 |

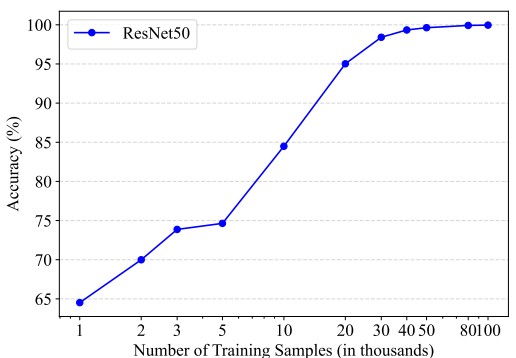

Figure 2: **Binary distribution classification on *label-to-image***. A positive correlation is observed between accuracy and the number of training samples. With only 50k samples per distribution, classifiers achieve over 99.5% accuracy.

Tab. 4. Notably, the self-supervised classifiers also demonstrate a strong ability to distinguish between real and generated distributions. Moreover, we observed two intriguing phenomena: First, unlike in semantic classification tasks on ImageNet, the masked image modeling-based MAE outperforms the contrastive learning-based MoCo v3 in identifying generated images related to ImageNet. Second, while self-supervised classifiers achieve relatively high accuracy on ImageNet-related tasks, their performance drops significantly when applied to CIFAR-10. We hypothesize this decline can be contributed to two factors: (1) MAE and MoCo v3 are pre-trained on ImageNet, not CIFAR-10, and (2) diffusion models perform exceptionally well on CIFAR-10 image generation, making it challenging for a simple linear classifier to differentiate between real and generated images. Besides, we also evaluate self-supervised classifiers in text-to-image scenarios (see detailed results in Sec. C.3).

> ***Conclusion* 1.** Despite the success of diffusion models in image generation, classifiers easily distinguish real from generated distribution, achieving high accuracy across various combinations. Besides, self-supervised classifiers also effectively identify generated distribution.

## 6 CLASSIFIER CONTRADICTIONS WITH FID AND HUMAN JUDGMENTS

Previously, we demonstrated that in most settings, classifiers can effortlessly distinguish between real and generated distributions, indicating that the generated distribution is still far from the real

Table 4: **Self-supervised classifiers demonstrate the ability to distinguish between real and generated distributions.** For each self-supervised method, we use ViT-B as the backbone.

| Real dataset | Generative model | Self-supervised method | Accuracy |
|---|---|---|---|
| *C* | *U* (Bao et al., 2023) | MAE | 71.83 |
| | | MoCo v3 | 73.39 |
| *C* | *E* (Karras et al., 2022) | MAE | 64.83 |
| | | MoCo v3 | 67.69 |
| *I*-256 | *U*-H/2 (Bao et al., 2023) | MAE | 91.65 |
| | | MoCo v3 | 77.38 |
| *I*-512 | *E*2-XXL (Karras et al., 2023) | MAE | 81.80 |
| | | MoCo v3 | 74.30 |

Table 5: **Distribution classification accuracy for various combinations of diffusion models with similar FIDs and ImageNet**. Notably, classifier accuracy improves as more distributions are added. Here, combinations refer to multiple distributions, where the number of distributions corresponds to the number of classes in distribution classification.

| Distribution combinations | Model | Accuracy |
|---|---|---|
| *U*-H/2, *D* | ResNet-50 | 99.66 |
| | ConvNeXt-T | 97.59 |
| | ViT-S | 86.13 |
| *U*-H/2, *D*, *I*-256 | ResNet-50 | 99.87 |
| | ConvNeXt-T | 99.77 |
| | ViT-S | 95.90 |
| *U*-H/2, *D*, *I*-256, *E*2-XXL | ResNet-50 | 99.91 |
| | ConvNeXt-T | 99.92 |
| | ViT-S | 98.13 |

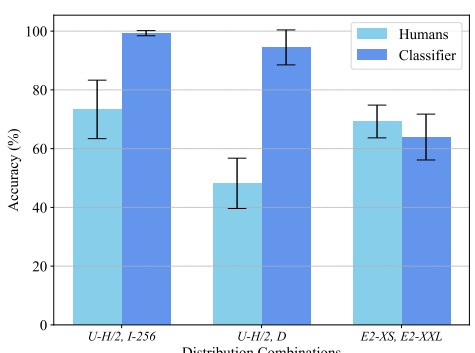

Figure 3: **User study.** Classifiers can easily distinguish between diffusion models with similar FIDs (U-H/2, D) but struggle with models from the same family that differ in parameters (E2-XS, E2-XXL). In contrast, humans show the opposite trend.

one. In this section, we explore the intriguing contradictions between classifiers and widely used evaluation protocols, such as FID and human judgments, through two sets of experiments. The first experiment compares EDM2-XS and EDM2-XXL, which share the same network architectures and training methodologies but differ in the number of parameters, resulting in FID scores of 2.91 and 1.81, respectively. The second experiment examines DiT-XL, U-ViT-H/2, and EDM2-XXL, which, despite having similar FIDs, vary in network architecture, sampling methods, and training protocols.

To analyze human judgments, we conducted a user study to evaluate the human ability to distinguish between different distributions (see the user interface in Fig. 10). Further details of the study are provided in Appendix C.6. The results are shown in Fig. 3, where individual variability is represented using error bars, indicating the standard deviation across participants/classifier performance.

In our first set of experiments, we analyze models within the same family: EDM2-XS and EDM2-XXL, the smallest and largest models in the EDM2 family, respectively. As shown in Tab. 6, we report the distribution classification accuracy for distinguishing between EDM2-XS and EDM2-XXL. The results show that classifiers struggle to differentiate between models from the same family. Only ResNet-50 shows some ability to distinguish them, while the other classifiers perform close to random chance. This contrasts sharply with the task of distinguishing between ImageNet-512 and EDM2-XXL (in Tab. 2), where all classifiers achieve over 86% accuracy. Thus, compared to distinguishing between diffusion models and the real distribution, neural networks view models within the same family—due to their shared inductive biases—as learning similar distributions, making them difficult to distinguish. Furthermore, in our user study (see Fig. 3), unlike the other tasks where classifiers significantly outperform humans, in the task of distinguishing between models within the same family, humans perform comparably to the classifiers. This observation highlights a limitation of neural

Table 6: **Classifiers struggle to differentiate between the smallest and largest EDM2 models.** Despite notable differences in FID (E2-XS: 2.91, E2-XXL:1.81), the classifiers show limited ability to accurately distinguish between these EDM2 variants.

| Combinations | Classifier | Accuracy (%) |
|---|---|---|
| E2-XS, E2-XXL | ResNet-50 | 74.91 |
| | ConvNeXt-T | 57.35 |
| | ViT-S | 59.57 |

Table 7: **Classifiers achieve high accuracy across crop sizes**. $^\dagger$ denotes center crop. $\star$ denotes random crop, where results are averaged over four trials.

| Combinations | Crop size | Accuracy (%) |
|---|---|---|
| U-H/2, I-256 | None | 99.96 |
| | $128^\dagger$ | 99.96 |
| | $64^\dagger$ | 99.97 |
| | $32^\dagger$ | 99.97 |
| | $16^\dagger$ | 99.78 |
| | $128^\star$ | 99.97 |
| | $64^\star$ | 99.97 |

network-based classifiers: despite the noticeable difference in FID scores between EDM2-XS and EDM2-XXL (2.91 and 1.81, respectively), the classifiers struggle to differentiate between them.

In our second group of experiments, we analyze different diffusion models with similar FIDs. As shown in Tab. 5, we present the distribution classification accuracy for three combinations of diffusion models with similar FIDs. We observe that classifiers can easily distinguish between these models, achieving extremely high accuracy, while humans perform close to random guessing (see Fig. 3). This highlights a key advantage of classifiers over widely used evaluation protocols: despite the FID scores being very close and humans being unable to differentiate between the models, classifiers can easily identify the differences. Moreover, an intriguing phenomenon occurs when additional distributions are introduced: the classifier's accuracy improves further. This contrasts with the findings in Liu & He (2024). A possible explanation is that all of these distributions are either directly related to ImageNet or generated by diffusion models trained on it. As new distributions are added, the classifier can leverage newly learned patterns to enhance the classification accuracy for previous combinations.

> ***Conclusion 2.*** Classifiers conflict with FID and human assessments in distribution classification. They can distinguish between different diffusion models with similar FID but struggle to do so within the same family. In contrast, humans show a different trend.

## 7 IMPLICATIONS OF THE DISTRIBUTION CLASSIFICATION TASK

Previously, we demonstrated the classifier's ability to distinguish real from generated distributions, as well as the contradictions between classifiers and evaluation methods like FID and human judgments. In this section, we show some important implications of the distribution classification task.

### 7.1 DIAGNOSING THE PROBLEM WITHIN DIFFUSION MODELS

A natural implication of classifiers is to diagnose the problem within diffusion models. Our approach complements traditional metrics like FID and human judgments by offering a more challenging evaluation method. It can also be extended to analyze specific features of generated data, providing deeper insights into the intrinsic characteristics of diffusion models, such as spatial and frequency features. In this section, we focus on advanced diffusion transformers, specifically U-ViT-H/2 (Bao et al., 2023), and investigate two key aspects: spatial features and frequency ranges. Our objective is to identify which spatial features diffusion models learn well, making them harder to distinguish, and which they struggle with, making them easier to detect. Additionally, we assess performance across different frequency ranges to pinpoint where diffusion models succeed and where they fail. For this analysis, we use U-ViT-H/2 trained on ImageNet-256, with ResNet-50 as the classifier.

In our first set of experiments, we focus on the spatial features. We start by progressively reducing central information through center-cropping images from a resolution of 256 down to 128, 64, 32, and 16, followed by training the classifier from scratch. The results, shown in Tab. 7, indicate that even with minimal central information, the classifier can accurately distinguish between real and generated distributions. We then assess whether classifiers can differentiate based on any arbitrary part of the image. We cropped images at resolutions of 128 and 64, and randomly selected four

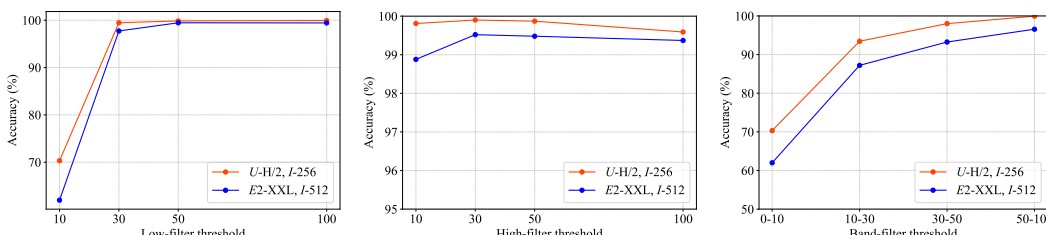

(a) Accuracy vs. low-frequency filter threshold

(b) Accuracy vs. high-frequency filter threshold

(c) Accuracy vs. band-frequency filter threshold

Figure 4: **Classifier achieves high accuracy across various combinations with limited frequency components,** except when only minimal low-frequency components are present (in subfig (a) low-pass filter threshold = 10, in subfig (c) band-pass filter threshold: 0-10).

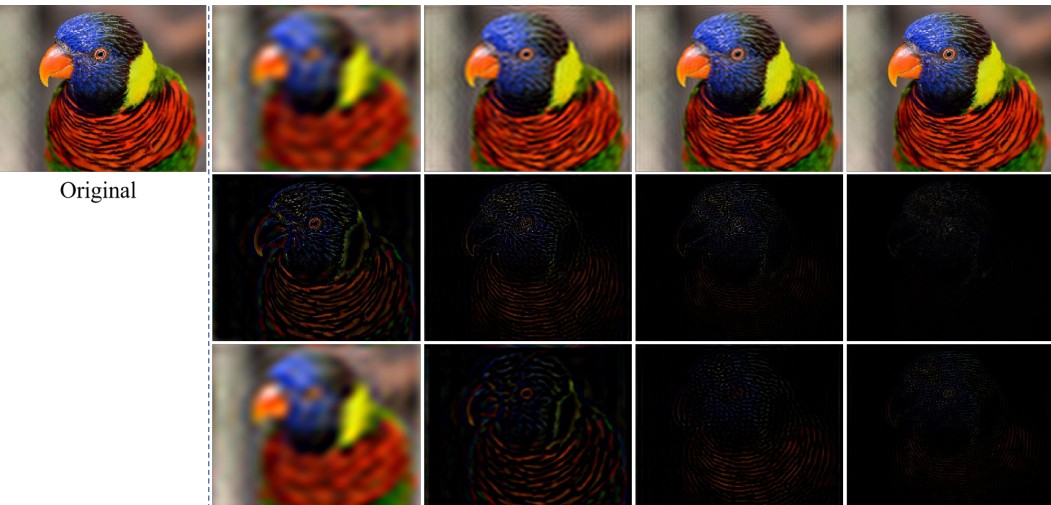

Figure 5: **Visualization of frequency domain processing.** *Top*: Low-pass filters, *Middle*: High-pass filters, each processed with increasing thresholds: 10, 30, 50, 100. *Bottom*: Band-pass filters, processed with band thresholds: 0-10, 10-30, 30-50, 50-100.

sections per image for training (see Fig. 7 for visualizations). The classifiers achieved accuracies of $99.97 \pm 0.004\%$ for 128 and $99.97 \pm 0.007\%$ for 64 (see Tab. 7). These results demonstrate that even with the advanced diffusion transformer U-ViT-H/2, which achieves an FID of 2.29, classifiers can reliably distinguish between real and generated images using any spatial part of the image.

In our second set of experiments, we focus on the frequency aspect. We preprocess images using low-pass, high-pass, and band-pass filters as described in Sec. 2.2. For the low-pass and high-pass filters, we apply thresholds of 10, 30, 50, and 100, while for the band-pass filter, we use frequency intervals of 0-10, 10-30, 30-50, and 50-100. Fig. 5 provides examples of the original and filtered images. As shown in Fig. 4, using ResNet-50 as the classifier, we observe that classifiers struggle to maintain high accuracy when low-frequency information is minimal but perform well across all other frequency intervals, even with only high-frequency components (e.g., threshold of 100). These results suggest that diffusion models effectively learn low-frequency features, making it more difficult for classifiers to distinguish real from generated images in this range. However, in higher frequency bands, diffusion models perform worse, making it easier for classifiers to identify generated images.

## 7.2 EVALUATING THE USE OF GENERATED DATA

Another natural implication is using classifiers to evaluate the use of generated data. Our previous findings show a significant difference between generated and real distributions, raising the question of whether this mismatch causes undesirable behaviors in models trained on synthetic data. One

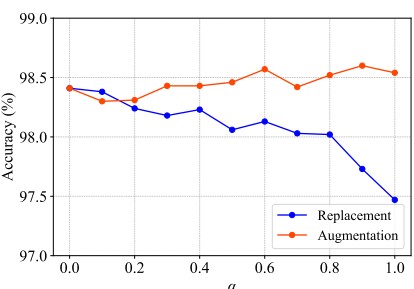

Figure 6: **Top-1 accuracy on the CIFAR-10 test set: Replacement vs. Augmentation.** $\alpha$ represents the ratio of EDM_S mixed with CIFAR-10.

Table 8: **Augmenting real data with generated data is more effective than replacing it.** $\star$ indicates results from training with CIFAR-10 (real data); $\dagger$ and $\ddagger$ indicate results where CIFAR-10 is replaced or augmented with EDM_S, respectively.

| Method given # labels per class (label fraction) | Error rate ↓ 4 (0.08%) |
|---|---|
| FlexMatch[*](Zhang et al., 2021) | $4.97_{\pm0.06}$ |
| FlexMatch[†] | $5.85_{\pm0.02}$ |
| FreeMatch[*](Wang et al., 2023a) | $4.90_{\pm0.04}$ |
| FreeMatch[†] | $5.94_{\pm0.10}$ |
| FreeMatch[‡](You et al., 2024) | $4.68_{\pm0.17}$ |

well-known phenomenon is that training generative models on data produced by previous generations in a continuous loop leads to a gradual decline in output quality and diversity (Shumailov et al., 2024; Alemohammad et al., 2024), ultimately resulting in model collapse, known as Model Autophagy Disorder (MAD). This is surprising, as data generated by modern diffusion models often exhibit high realism. However, using this data to train next-generation models triggers MAD. Our approach provides insight into this issue: although these models generate realistic images, the distributions they produce remain distinct from the real distribution—especially when viewed by classifiers—leading to a distribution mismatch. As this mismatch accumulates, the generated distribution drifts further from the original. To address this, Alemohammad et al. (2024) propose incorporating fresh real samples into the training data for each generation, rather than relying solely on generated data.

To better understand this phenomenon, we further explore the impact of using generative model data in downstream tasks. Specifically, we investigate whether the distribution mismatch between real and generated data causes undesirable behaviors when replacing real data with generated data in tasks such as classification. To analyze this, we use EDM (Karras et al., 2022) to generate the same number of samples as CIFAR-10, creating a synthetic dataset, EDM_S, and perform both supervised and semi-supervised classification to assess the impact. For supervised classification, we train the classifier with ConvNeXt settings (Liu et al., 2022) for CIFAR-10, using the original test set for evaluation. To create the training dataset, we mix EDM_S and CIFAR-10 with a ratio $\alpha$. We employ two strategies: one replaces $\alpha$ of CIFAR-10 with EDM_S, where $\alpha$ represents the replacement ratio, and the other augments EDM_S to CIFAR-10. As shown in Fig. 6, we observe that due to the distribution mismatch, simply replacing samples leads to a decline in accuracy. As $\alpha$ increases from 0 to 1, the top-1 accuracy gradually declines. In contrast, when augmenting EDM_S to CIFAR-10, accuracy generally improves for most $\alpha$ values. For semi-supervised classification, we use FreeMatch (Wang et al., 2023a) and FlexMatch (Zhang et al., 2021). As shown in Tab. 8, simply replacing CIFAR-10 with EDM_S reduces performance: FlexMatch's error rate increases from $4.97_{\pm0.06}$% to $5.85_{\pm0.02}$%, and FreeMatch's from $4.90_{\pm0.04}$% to $5.94_{\pm0.10}$%, consistent with the results in supervised classification. Meanwhile, DPT (You et al., 2024) augments CIFAR-10 with EDM-generated data, training with FreeMatch, and achieves a lower error rate of $4.68_{\pm0.17}$%.

From the analysis above, we observe that the distribution mismatch between generated and real data leads to undesirable behaviors, such as "MAD" or performance degradation, when real data is simply replaced with generated data. However, augmenting real data with generated data is more effective.

## 8 CONCLUSION

Our paper proposes the distribution classification tasks to explore a fundamental question: *How far is the generated distribution from the real distribution?* Our findings show that classifiers can effectively distinguish between diffusion-generated and real distributions, even when diffusion models achieve low FID scores and generate lifelike images. Additionally, our study shows contradictions between classifier performance and evaluation metrics such as FID and human judgments. Finally, our methodology can naturally serve as a diagnostic tool for diffusion models, offering a reasonable explanation for Model Autophagy Disorder (MAD) and evaluating the use of generated data.

ETHICS STATEMENT

This paper enhances the understanding of generated data and the differences between generated and real distributions, including shedding light on the model autophagy disorder phenomenon and evaluating the use of generated data. However, our findings could be used to improve the realism of generated images, potentially making "DeepFakes" more difficult to detect if misused. This issue can be mitigated by advancing automatic detection methods, which is an active area of research.

REPRODUCIBILITY STATEMENT

Our submission includes PyTorch code to ensure research reproducibility. Please refer to `README.md` for detailed instructions such as setting up the Anaconda environment. The datasets used are public and detailed in Appendix A. Pretrained models and diffusion sampling implementations are accessible via the code links listed in Table 9, including scripts for label-to-image generation (e.g., U-ViT, DiT, EDM) and text-to-image generation (e.g., Pixart-$\alpha$, SDXL, Playground-v2.5).

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

Table 9: **Code links and licenses.**

| Method | Link | License |
|---|---|---|
| ConvNeXt | `https://github.com/facebookresearch/ConvNeXt` | MIT License |
| U-ViT | `https://github.com/baofff/U-ViT` | MIT License |
| DiT | `https://github.com/facebookresearch/DiT` | CC BY-NC 4.0 |
| EDM | `https://github.com/NVlabs/edm` | CC BY-NC-SA 4.0 |
| EDM2 | `https://github.com/NVlabs/edm2` | CC BY-NC-SA 4.0 |

| Model | Link (add 'https://huggingface.co/') | License |
|---|---|---|
| PixArt-$\alpha$ | `PixArt-alpha/PixArt-XL-2-1024-MS` | Open RAIL++-M |
| SDXL | `stabilityai/stable-diffusion-xl-base-1.0` | Open RAIL++-M |
| Playground-v2.5 | `playgroundai/playground-v2.5-1024px-aesthetic` | Playground v2.5 |

# A   DETAIL EXPERIMENT SETTINGS

We implement our experiments upon the official code of ConvNeXt (Liu et al., 2022), U-ViT (Bao et al., 2023), DiT (Peebles & Xie, 2023), EDM (Karras et al., 2022), EDM2 (Karras et al., 2023). We also utilize the official models of Pixart-$\alpha$ (Chen et al., 2023), SDXL (Podell et al., 2023), and Playground-v2.5 (Li et al., 2024). The respective links and licenses are detailed in Tab. 9.

We present the main experiment settings as follows.

**Dataset.** For label-to-image, we consider the CIFAR (Krizhevsky et al., 2009) and ImageNet (Deng et al., 2009) datasets, which are well-established and widely recognized benchmarks in the field of image generation. For text-to-image, we utilize the COCO2014 dataset (Lin et al., 2014), known for its rich annotations and diverse image content. For CIFAR-10, we use the real CIFAR-10 training set and validation set to construct the real distribution and use the diffusion model to generate 50k training images and 10k validation images to construct the generated distribution. In the case of ImageNet, we consider 256 and 512 resolutions, which are common resolutions for ImageNet image generation tasks. We randomly sample 100k training images and 50k validation images from the training set and the validation set of ImageNet to construct the real distribution and use the diffusion models to generate 100k training images and 50k validation images to construct the generated distribution. For the real dataset, we adopt the data processing method from ADM (Dhariwal & Nichol, 2021) to modify the ImageNet dataset into two common resolutions: ImageNet-256 and ImageNet-512, where the numbers indicate the resolution of the data. In addition, for the COCO2014 dataset, by default, we construct the real distribution by randomly sampling 10k training images and 1k validation images from the respective training and validation sets of COCO2014. Each image in this dataset is associated with five captions. To create the generated distribution, we randomly select one of the five captions for each real image to serve as a prompt. These prompts are then used by the diffusion models to generate an equivalent number of images to construct the generated distribution.

**Classifier.** By default, we employ the ResNet-50 (He et al., 2016) as the classifier architecture. For completeness, we also consider ConvNeXt-T (Liu et al., 2022) and ViT-S (Dosovitskiy et al., 2020). Our pre-processing protocol follows the standard supervised training approach (Liu et al., 2022). Specifically, during training, classifiers process randomly augmented crops of $224 \times 224$ images. During validation, images are resized so that their smaller dimension reaches 256 pixels while preserving the original aspect ratio. Subsequently, these images are center cropped to $224 \times 224$ pixels before being fed into the model. For the experiments on CIFAR-10, we initialize our classifier with the ResNet-50, pre-trained on ImageNet. For completeness, we also present the result trained from scratch (see experiments in Appendix. C.2). In the case of ImageNet, we opt to train the ResNet-50 model from scratch.

**Diffusion model.** For the generation of CIFAR-10, we consider two diffusion models: EDM (Karras et al., 2022) and U-ViT (Bao et al., 2023). Both models have demonstrated strong generation performance on the CIFAR-10 (Krizhevsky et al., 2009). Quantitatively, EDM achieves an FID of 1.79, and U-ViT achieves an FID of 3.11. To improve efficiency, we modified U-ViT's sampling method from Euler-Maruyama to DPM-Solver (Lu et al., 2022) and reduced the sampling steps from 1,000 to 50. These adjustments resulted in U-ViT achieving an FID of 3.65 on CIFAR-10. For

Table 10: **Training settings for CIFAR-10.**

| Config | Value |
|---|---|
| Optimizer | AdamW |
| Learning rate | 4e-4 |
| Weight decay | 0.05 |
| Optimizer momentum | $\beta_1, \beta_2$=0.9, 0.999 |
| Batch size | 256 |
| Learning rate schedule | Cosine decay |
| Warmup epochs | 0 |
| Training epochs | 50 |
| Augmentation | RandAug (9, 0.5) |
| Label smoothing | 0.1 |
| Mixup | 0 |
| Cutmix | 0 |

Table 11: **Training settings for ImageNet.**

| Config | Value |
|---|---|
| Optimizer | AdamW |
| Learning rate | 1e-3 |
| Weight decay | 0.3 |
| Optimizer momentum | $\beta_1, \beta_2$=0.9, 0.95 |
| Batch size | 4096 |
| Learning rate schedule | Cosine decay |
| Warmup epochs | 20 |
| Training epochs | 200 |
| Augmentation | RandAug (9, 0.5) |
| Label smoothing | 0.1 |
| Mixup | 0.8 |
| Cutmix | 1.0 |

Table 12: **Training time of classifiers.**

| Model | Combinations | Epochs | GPU-type | GPU-nums | Hours |
|---|---|---|---|---|---|
| ResNet-50 | $C, U$ | 50 | 3090 | 4 | 2 |
| ViT-S | $C, U$ | 50 | 3090 | 4 | 2 |
| ConvNeXt-T | $C, U$ | 50 | 3090 | 4 | 2 |
| ResNet-50 | $I$-256, $U$-H/2 | 200 | V100 | 8 | 6 |
| ViT-S | $I$-256, $U$-H/2 | 200 | V100 | 8 | 9 |
| ConvNeXt-T | $I$-256, $U$-H/2 | 200 | V100 | 8 | 8 |
| ResNet-50 | $I$-512, $E2$-XXL | 200 | V100 | 8 | 9 |
| ViT-S | $I$-512, $E2$-XXL | 200 | V100 | 8 | 11 |
| ConvNeXt-T | $I$-512, $E2$-XXL | 200 | V100 | 8 | 10 |

the generation of ImageNet-256, we explore three diffusion models: EDM2 (Karras et al., 2023), U-ViT-H/2 (Bao et al., 2023), DiT-XL (Peebles & Xie, 2023). Both DiT and U-ViT are prominent diffusion transformer architectures known for their scalability and strong performance. U-ViT-H/2 achieves an FID of 2.29 on ImageNet-256, and DiT-XL/2 achieves an FID of 2.27. We consider EDM2 to incorporate a UNet-based architecture, which was traditionally used before the rise of diffusion transformers. Since EDM2 is originally designed for ImageNet-512 generation, we resize the generated images from 512 to 256 resolution to suit our ImageNet-256 task. In this way, EDM2-XXL achieves an FID of 2.14 on this task, which is similar to the FID achieved by U-ViT and DiT. For the generation of ImageNet-512, we use EDM2 (Karras et al., 2023), which achieves state-of-the-art performance on this task with an FID of 1.81. For the generation of COCO, we consider three state-of-the-art text-to-image diffusion models: Pixart-$\alpha$ (Chen et al., 2023), SDXL (Podell et al., 2023), and Playground-v2.5 (Li et al., 2024).

**Evaluation.** We use the top-1 accuracy on the validation set to evaluate classification performance.

**Training settings.** The complete training settings of ResNet-50 are reported in Tab. 10 for combinations related to CIFAR-10 and Tab. 11 for combinations related to ImageNet.

# B    COMPUTATIONAL COST

Our experiments were conducted on RTX 3090 and V100 GPUs. The detailed computational costs are presented in Tab. 12. Training epochs were set to 50 for CIFAR-10 and 200 for ImageNet. The number of epochs trained on CIFAR-10 is relatively low because we use a pre-trained model to initialize our classifier, enabling faster convergence.

Table 13: **Distribution classification accuracy on CIFAR-10.** "Pretrained" indicates that the classifier was initialized with a model pretrained on ImageNet, while "Scratch" indicates that the classifier was trained from scratch.

| Real dataset | Generative model | Model | Pretrained | Scratch |
|---|---|---|---|---|
| $C$ | $E$ (Karras et al., 2022) | ResNet-50 | 96.25 | 56.13 |
| | | ViT-S | 89.38 | 55.32 |
| | | ConvNeXt-T | 98.43 | 53.27 |
| $C$ | $U$ (Bao et al., 2023) | ResNet-50 | 99.92 | 56.70 |
| | | ViT-S | 98.04 | 52.66 |
| | | ConvNeXt-T | 99.96 | 56.37 |

Table 14: **Binary distribution classification on label-to-image**. All classifiers yield high accuracy on various datasets against strong generative models. FIDs are taken from the corresponding references.

| Real dataset | Generative model | FID | Classifier | Accuracy (%) |
|---|---|---|---|---|
| $I$-256 | StyleGAN-XL (Sauer et al., 2022) | 2.30 | ResNet-50 | 99.69 |
| | | | ViT-S | 99.95 |
| | | | ConvNeXt-T | 96.85 |

## C   ADDITIONAL RESULTS

### C.1   ADDITIONAL RESULTS FROM GENERATIVE ADVERSARIAL NETWORK

As shown in Tab. 14, we conducted experiments using StyleGAN-XL (Sauer et al., 2022), a state-of-the-art GAN model trained on ImageNet-256. Our results demonstrate that, despite the use of a discriminator during GAN training, the classifier can still easily distinguish between real and generated images. We argue that this is because the discriminator is trained jointly with the generator. During training, the generated data seen by the discriminator comes from a continuously evolving distribution, as the generator improves with each iteration. However, when using a classifier to distinguish between real and generated distributions, the generated distribution remains fixed.

### C.2   RESULTS OF CIFAR-10

In order to ensure the completeness of the experiment, we are here to present the result of CIFAR-10 trained from scratch. We present the results in Tab. 13. If there is no prior knowledge, classifiers struggle to distinguish between real and generated data in datasets with low resolution, such as CIFAR-10 (i.e., 32x32). However, the use of a pre-trained model allows the features learned from the ImageNet dataset to aid in differentiating between real and generated images in CIFAR-10.

### C.3   SELF-SUPERVISED CLASSIFIERS FOR TEXT-TO-IMAGE DISTRIBUTION CLASSIFICATION

As shown in Tab. 15, we report the distribution classification accuracy of self-supervised classifiers in text-to-image scenarios. Notably, these classifiers achieve high accuracy in distinguishing between different text-to-image models. This suggests that there are significant differences between text-to-image generative models, allowing even self-supervised classifiers to easily distinguish them.

### C.4   VISUALIZATION OF CROPS

As shown in Fig. 7, we present the visualization of crops mentioned in Sec. 7.1.

### C.5   FREQUENCY ANALYSIS ON THE COMBINATION OF EDM2-XS AND EDM2-XXL

We preprocess the generated images using band-pass filters as defined in Sec. 2.2, with four threshold intervals: 0-10, 10-30, 30-50, and 50-100. An example of original and processed EDM2-XXL

Table 15: **Four-way distribution classification on text-to-image**. Linear classifier on self-supervised features also achieves high accuracy in distinguishing four distributions: COCO (Lin et al., 2014), Pixart-$\alpha$ (Chen et al., 2023), SDXL (Podell et al., 2023), and Playground-v2.5 (Li et al., 2024), using only 5k or 10k training samples per dataset. We use ViT-B as the backbone.

| Training samples | Self-supervised method | Accuracy (%) |
|---|---|---|
| 5k | MAE | 87.77 |
| 5k | MoCo v3 | 90.48 |
| 10k | MAE | 91.67 |
| 10k | MoCo v3 | 90.90 |

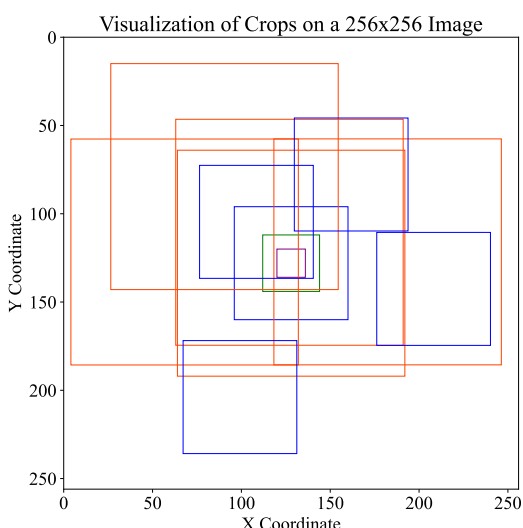

Figure 7: **Visualization of crops.** With each resolution represented by a different color.

generated images is shown in Fig. 8. As shown in Tab. 16 and Fig. 9, for EDM2-XS and EDM2-XXL, the smallest and largest models in the EDM2 family, classifiers' accuracy approaches random guessing (around 50%) across different threshold intervals. This indicates that for models within the same diffusion model family, which share inductive biases but differ in visual quality, classifiers unable to distinguish between them based on any specific frequency band.

C.6 USER STUDY

Fig. 10 shows a screenshot of the interaction interface used in our user study. The study involved nineteen participants, and we designed three groups of experiments, each requiring participants to classify 32 pairs of images. In the first set, participants distinguished between generated and real images, with real images sourced from ImageNet-256 and generated images from U-ViT-H/2. In the

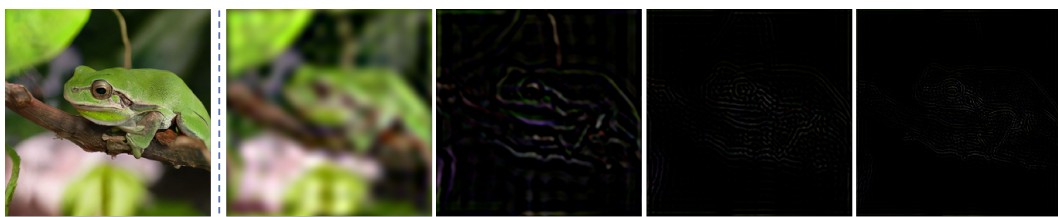

Figure 8: **Visualization of frequency domain processing of EDM2-XXL.** The image shows the results after applying a band-pass filter to EDM2-XXL with band thresholds of 0-10, 10-30, 30-50, and 50-100, from left to right.

Table 16: **Classification accuracy** of ResNet-50 on the combination of EDM2-XS and EDM2-XXL after applying band-pass filters.

| Combinations | Classifier | Threshold Intervals | Accuracy |
|---|---|---|---|
| $E2$-XS, $E2$-XXL | ResNet-50 | 0-10 | 57.94 |
| | | 10-30 | 58.96 |
| | | 30-50 | 58.12 |
| | | 50-100 | 50.48 |

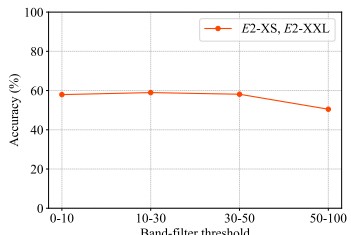

Figure 9: **Accuracy vs. band-frequency filter threshold**

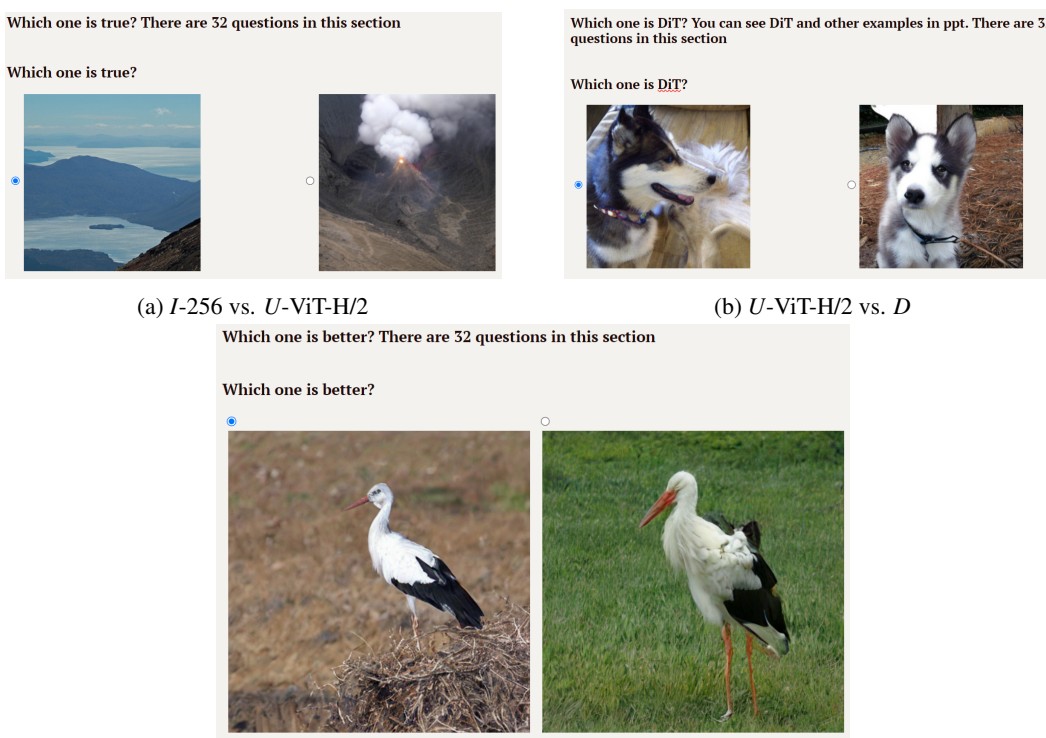

(a) *I*-256 vs. *U*-ViT-H/2

(b) *U*-ViT-H/2 vs. *D*

(c) *E*2-XS vs. *E*2-XXL

Figure 10: **Screenshot of user study.** Participants are asked to distinguish generated distributions from real ones and to classify which diffusion model generated a given image. Each group of experiments is illustrated with a separate example here. In each set of experiments, we also randomized the order to prevent examples from the same set from influencing each other.

second set, they classified images between two diffusion models with similar performance: DiT-XL/2 and U-ViT-H/2. For the final set, participants evaluated images from EDM2-XS and EDM2-XXL, which share the same training methodology but differ in parameter count, resulting in different FID scores and visual quality. In the first set, participants were asked to identify the real images. In the second, they were tasked with identifying DiT images, and reference images from DiT and U-ViT were provided during the test. In the third experiment, participants judged which images were of higher quality. All nineteen participants were graduate students with substantial experience in machine learning. They were allowed to zoom in on the images during the experiment, which was conducted on 27-inch 4K displays. All participants had corrected vision of 1.0 (standard normal vision), and their ages ranged from 22 to 26. Each participant completed the experiment within an hour and was compensated $10.

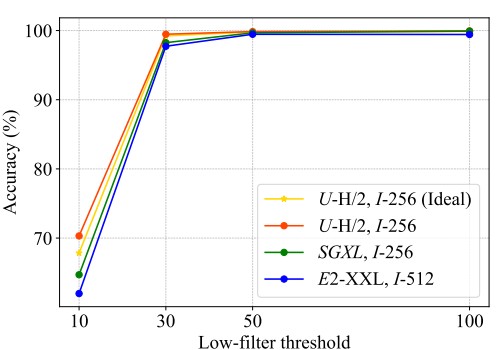
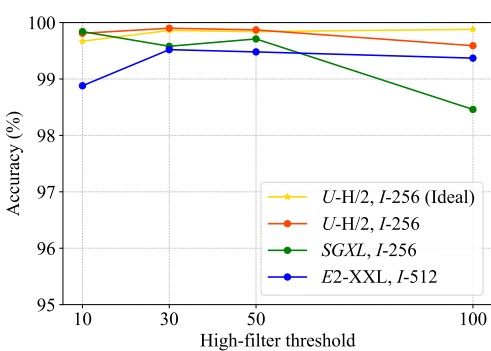

(a) Accuracy vs. low-frequency filter threshold

(b) Accuracy vs. high-frequency filter threshold

Figure 11: Comparison of model accuracy across different filter thresholds using rectangular and circular filters.

## D  BINARY CLASSIFICATION AS A MEASURE OF DISTRIBUTION DISTANCE

We employ a classifier $C(x)$ to distinguish between the real data distribution $p_{\text{data}}(x)$ and the generated data distribution $p_g(x)$. By training $C(x)$ using the binary cross-entropy loss:

$$L(C) = -E_{x \sim p_{\text{data}}(x)}[\log(C(x))] - E_{x \sim p_g(x)}[\log(1 - C(x))], \tag{5}$$

The optimal classifier that minimizes this loss is:

$$C^*(x) = \frac{p_{\text{data}}(x)}{p_{\text{data}}(x) + p_g(x)}. \tag{6}$$

Substituting $C^*(x)$ back into the loss function yields:

$$L(C^*) = -\log(4) + 2\,\text{JSD}(p_{\text{data}}(x) \| p_g(x)), \tag{7}$$

where JSD denotes the Jensen-Shannon Divergence—a direct measure of the distance between the two distributions.

In our paper, we use classification accuracy to evaluate how well the classifier distinguishes between real and generated data because it provides an intuitive and interpretable metric. Although accuracy is non-differentiable and unsuitable for direct optimization, training the classifier with the binary cross-entropy loss—a convex surrogate—often leads to improved accuracy. This correlation suggests that accuracy can serve as a proxy for changes in the loss function, reflecting the distance between the generated and real data distributions.

## E  RELATIONSHIP BETWEEN DISTRIBUTION CLASSIFICATION AND FID

The Fréchet Inception Distance (FID) is a widely used metric for evaluating the quality of generative models. It relies on feature extraction networks trained on datasets such as ImageNet and assumes that the extracted feature vectors follow a multivariate Gaussian distribution. FID calculates the Fréchet distance between these Gaussian distributions to measure discrepancies between the real and generated data. Meanwhile, as noted by Kynkäänniemi et al. (2022), FID can decrease simply by aligning the histograms of top-N classifications, without necessarily improving the perceptual quality of the generated images. Additionally, recent works like Karras et al. (2023) and Tian et al. (2024) report FID scores close to those of the ImageNet validation set, suggesting limitations in FID's sensitivity to certain distribution differences.

In contrast, our classifier-based approach offers a more direct measure of the distance between distributions without relying on the Gaussian assumptions inherent in FID. By training a classifier to distinguish between real and generated data, we obtain an intuitive and interpretable metric that reflects the actual distribution differences. This method complements commonly used metrics such as FID and Inception Score (IS), providing an alternative perspective on evaluating generative models.

# F  COMPARISON OF RECTANGULAR AND CIRCULAR FILTERS

Rectangular and circular filters are common techniques in image filtering. In this paper, we chose to implement a rectangular filter following the official FreeU implementation (Si et al., 2024), due to its simplicity and computational efficiency. For comparison, Fig. 11 presents initial results using a circular mask, denoted as U-H/2 and I-256 (Ideal). The results from both implementations are similar, and we chose to use the rectangular filter for its computational efficiency.

