# OpenReview forum: "Are Images Indistinguishable to Humans Also Indistinguishable to Classifiers?"
_ICLR.cc/2025/Conference — ICLR 2025 Conference Withdrawn Submission_

### Official Review · Reviewer_TUyF · 2024-10-29

**Soundness:** 3
**Presentation:** 2
**Contribution:** 2
**Rating:** 5
**Confidence:** 3

**Summary:**

This paper aims to explore the distributional differences between real and high-quality generated data, mainly focusing on diffusion models. By training neural network classifiers to differentiate real from generated distributions, the authors found that classifiers could easily identify generated data, even when human perception and FID scores suggest high-quality results. The study also looks into distributional discrepancies across model types, revealing that classifiers can distinguish between models with similar FID scores from different families but struggle with models of varying capacities and FID scores within the same family, which again shows contradictory results between FID and the classifiers. Additionally, the authors analyze which features classifiers emphasize, finding a reliance on high-frequency features and spatial information to distinguish distributions. The paper further connects these observations to the Model Autophagy Disorder (MAD) phenomenon, offering broader implications for understanding distributional gaps between real and generated data. Overall, the paper aims to diagnose the differences between real and generated images by leveraging classifier-based analysis.

**Strengths:**

1. The paper addresses a highly relevant issue: exploring the differences between real and generated data, which is crucial given the advancements in generative AI.
1. The paper has interesting findings by comparing generated data between different diffusion models: classifiers distinguishing between models with similar FID scores from different families but struggling with models of different capacities within the same family. This insight suggests that FID scores may not capture all dimensions of model quality.
1. The paper makes solid connections to existing research, especially by relating its findings to the Model Autophagy Disorder (MAD) phenomenon, enhancing the broader significance of its results.
1. The author extended the analysis beyond diffusion model by including some results for GAN models, showing consistency in findings across model types. This cross-model analysis supports the generalizability of the conclusions.
1. Implementation details are well-documented, which strengthens reproducibility and provides clarity for readers who may want to replicate or build on the study.

**Weaknesses:**

My main concern with the paper is that while the paper includes a substantial amount of data and findings, it lacks a cohesive narrative that ties the results together and clarifies their implications.
1. For example, the paper demonstrates differences between FID and classifier results (sections 5.2 and 6) without in-depth discussions on the reasons behind these discrepancies. Section 7.1 shows some analysis on what the classifiers focus on, it would be helpful to perform similar analysis for the FID metrics, and pointing out the differences in what aspects of the images the two methods focus on.
1. Similarly, observations like the classifier’s poor performance with CIFAR data are presented without verification of potential causes (section 5.3). Testing with CIFAR10-pretrained self-supervised classifiers would strengthen these claims. For example, are diffusion models really closing the gap between real and generated images on low-resolution datasets?
1. Additionally, the purpose of using self-supervised models in classification (section 5.3) is unclear, and further context would help clarify their role.

Other concerns:
1. The analysis of models within the same family but with different capacities is limited to only one model (EDM2-XS and EDM2-XXL), which weakens support for this conclusion. Including additional models of varying capacities would provide more evidence.
1. The user study involves only 19 participants, which could introduce noise or bias. More information on participant agreement rates/variance of the results would enhance the reliability of these results.

**Questions:**

1. Could you expand on the reasons behind the observed discrepancies between FID and classifier results (sections 5.2 and 6)?

1. In section 5.3, would you consider testing CIFAR10-pretrained self-supervised classifiers to verify the hypothesis for the classifier’s poor performance with CIFAR data?

1. Could you clarify the purpose of using self-supervised models in classification in section 5.3?

1. What is the detailed experiment setup for Table 1? What is the training/testing data and label?

---

### Official Review · Reviewer_BdwV · 2024-10-29

**Soundness:** 2
**Presentation:** 3
**Contribution:** 2
**Rating:** 5
**Confidence:** 3

**Summary:**

This paper evaluates whether classifiers can reliably differentiate real images from generated images. Despite the visual similarity of these generated images to real ones, the study finds that classifiers achieve high accuracy in distinguishing between them. The research highlights several inconsistencies: classifiers can easily detect generated images that humans find indistinguishable and also reveal contradictions between classifier performance and widely used metrics like FID. The study concludes with practical insights, such as recommending data augmentation with generated data instead of full replacement to avoid issues like MAD during model training.

**Strengths:**

1.  This paper introduces classifier performance as a new way to evaluate generative models.
2. The study highlights contradictions between classifier performance, human judgment, and FID scores. Classifiers can easily distinguish between different diffusion models with similar FID but are challenging to differentiate between smaller and larger models from the same diffusion family.

**Weaknesses:**

1. The authors empirically discover interesting inconsistencies between classifier performance, human evaluations, and FID metrics but lack in-depth analysis of the underlying reasons. This results in relatively weak theoretical contribution.
2. While classifiers are introduced as a new evaluation method, the paper lacks a detailed discussion of where and why this approach is meaningful. The significance of using classifiers as a metric remains unclear.

**Questions:**

1. In scenarios where classifiers underperform compared to humans or FID (e.g., within the same model family), how can the classifiers be improved?
2. How are the human evaluation scores obtained? How to eliminate agreement bias among participants?

---

### Official Review · Reviewer_BTDj · 2024-11-02

**Soundness:** 3
**Presentation:** 3
**Contribution:** 2
**Rating:** 5
**Confidence:** 3

**Summary:**

This works experimentally researches the differences between the generated data and real data. The differences is evaluated by classifiers and the results reveal some surprising and interesting properties of generative models, which is hidden in the usual metric, such as human judgement and FID score. And these hidden properties are important for analyzing the generated data, like using them to train model.

**Strengths:**

- This work provides another view for analyzing the differences between the generated and real distributions, instead of just comparing FID. They propose a classifier-based approach to measure the differences, which is more useful for analyzing the results of generative models, especially for using the generated data for downstream tasks.

- Experimentally, this work finds contradictions between classifier's prediction and FID score, which is helpful for deeply analyzing the generated results of diffusion models. First, their approach shows the diffusion model can effectively learn specific features of data but ignore other features. Second, it provides a reasonable explanation of using generated data for training models.

- The experiments are solid and reasonable.

**Weaknesses:**

- This work has a lack of providing any theoretical analysis or theoretical intuitions to explain why the classifier, even self-supervised classifier can easily distinguish the generated data from real data, when human and FID score cannot see the differences.

- The authors may provide more details of FID metric and the proposed classifier-metric. They may explain the differences of the distances between the generated distribution and the real distribution provided by FID score and the distance provided by classifiers. Analyzing the differences may help us to understand the contradictions of FID score and classifiers for measuring the generated distributions.

**Questions:**

- In the part of researching the frequency aspect (line 472-480), do we need to compare the differences between the generated images and the real images after frequency filtering besides the accuracy provided by classifiers? Does it provide more solid evidence of the statement, diffusion models effectively learn low frequency components?

---

### Official Review · Reviewer_qfAs · 2024-11-04

**Soundness:** 2
**Presentation:** 3
**Contribution:** 2
**Rating:** 5
**Confidence:** 3

**Summary:**

The authors answer a fundamental question: How far is the generated distribution from the real distribution. They demonstrate that classifiers can distinguish between generated and real images with high accuracy—a distinction that humans are unable to make. Additionally, they provide analysis to diagnose the Model Attribution Discrepancy (MAD) phenomenon.

**Strengths:**

S1. The authors carried out studies under various relevant settings, e.g., classifier trained from scratch vs. SSL classifiers, etc.

S2. The authors made effective use of frequency filters, demonstrating their impact both qualitatively and quantitatively

S3. The diagnosis of MAD phenomenon offers a plausible explanation

**Weaknesses:**

W1. Prior work has shown the neural network-based classifiers are sensitive to high-frequency features/noises. Conducting a more in-depth analysis of the specific features that classifiers focus on could strengthen the paper, especially from an explainability perspective. For instance, the observation that samples from the same model family are harder to distinguish compared to samples from different model families raises the question of whether each model family has a unique “watermark” embedded in their generated images.

W2. Some related work should be discussed and compared, e.g., [C1] showed FID doesn't correlate well with human judgements, and [C2] on different metrics and low-level and high-level features.




[C1] Exposing flaws of generative model evaluation metrics and their unfair treatment of diffusion models, Stein et al., NeurIPS 2023

[C2] Dreamsim: Learning new dimensions of human visual similarity using synthetic data, Stephanie et al., NeurIPS 2023

**Questions:**

Q1. What is the temperature used for sampling, how will this affect the conclusion? For instance, more diverse generation vs. more duplicates.

Q2. In Tab.2, is transformer-based classifier worse than convolution-based classifier? Is this because the latter detects high-frequency features better.

Q3. How good is the classifier on CIFAR10 to do distribution classification?

Q4. Comparing Tab. 2 and Tab. 4, is it fair to say SSL models worse than classifiers train from scratch to classify distributions?

---

### Official Review · Reviewer_L7JN · 2024-11-04

**Soundness:** 3
**Presentation:** 2
**Contribution:** 1
**Rating:** 5
**Confidence:** 3

**Summary:**

This paper explores whether neural network-based classifiers can distinguish between real and generated images despite advances in diffusion models achieving realistic outputs and low Fréchet Inception Distance (FID) scores.

Through distribution classification tasks, the authors show that classifiers consistently and accurately differentiate between real and generated distributions, revealing a marked discrepancy that remains hidden in humans and FID scores. Additionally, classifiers can distinguish diffusion models of different architectures with similar FIDs, though they struggle with models of the same family but different scales. The study suggests that augmenting real data with generated images is more effective than replacing it entirely, as complete replacement causes distribution drift and performance degradation.

**Strengths:**

The authors propose a classifier-based approach to evaluating distribution differences, providing insights that complement traditional metrics like FID. This diagnostic tool can help understand diffusion model limitations beyond typical quality assessments.

The paper's findings regarding the distinctiveness of generated data, even when visually indistinguishable from real data, are relevant to the broader community working on generative models and dataset augmentation strategies.

The study includes experiments across multiple datasets (CIFAR-10, ImageNet, COCO) and models, providing a detailed comparison between real and generated distributions using various classifiers, architectures, and training samples. The results highlight classifier strength in identifying distribution discrepancies across these contexts.

**Weaknesses:**

1. One of my main concerns is that many of its findings seem to echo results from previous works, such as those cited in the related work section. Prior research has demonstrated that neural network classifiers can effectively distinguish real from generated images, which somewhat diminishes the unique contribution of this study. Without extending or innovating upon these prior insights, the paper's impact remains limited in originality.

2. The paper frequently refers to its approach as "distribution classification," implying a theoretical or practical connection to statistical concepts of distribution. However, the methodology appears identical to regular classification tasks focused on distinguishing real and generated images. This terminology may be misleading, as it does not introduce any specific statistical treatment or distributional modeling. Greater clarity around this term and why it differs from conventional classifier settings would be beneficial for readers.

3. The methodology mainly employs existing classifiers and conventional metrics, such as FID and classification accuracy, to evaluate generative models, leading to limited methodological novelty. The reliance on standard architectures (e.g., ResNet-50, ConvNeXt, ViT-S) and classification accuracy as a primary metric for assessing generative quality does not offer fresh insights or approaches to the problem. Coupled with the limited novelty in findings, this contributes to the perception that the paper lacks substantial innovation.

4. While the classifier-based evaluation offers diagnostic insights, the paper does not provide actionable improvements or enhancements for current generative models based on those insights.

**Questions:**

In addition to the weakness section:

1. The paper assumes high classification accuracy directly correlates with the quality of distribution differences, which may oversimplify the nuanced gaps in generated and real data. Further exploration or validation of why classifiers perform well on certain generative models would add depth to the analysis.

---

### Note · Authors · 2024-11-14

I have read and agree with the venue's withdrawal policy on behalf of myself and my co-authors.